# Abrocitinib Attenuates Microglia-Mediated Neuroinflammation after Traumatic Brain Injury via Inhibiting the JAK1/STAT1/NF-κB Pathway

**DOI:** 10.3390/cells11223588

**Published:** 2022-11-13

**Authors:** Tuo Li, Lei Li, Ruilong Peng, Hongying Hao, Hejun Zhang, Yalong Gao, Cong Wang, Fanjian Li, Xilei Liu, Fanglian Chen, Shu Zhang, Jianning Zhang

**Affiliations:** 1Department of Neurosurgery, Tianjin Medical University General Hospital, Tianjin 300000, China; 2Tianjin Neurological Institute, Tianjin 300000, China; 3Graduate School, Tianjin Medical University, Tianjin 300000, China; 4Key Laboratory of Post-Trauma Neuro-Repair and Regeneration in Central Nervous System, Ministry of Education, Tianjin 300000, China; 5Tianjin Key Laboratory of Injuries, Variations and Regeneration of Nervous System, Tianjin 300000, China; 6Department of Neurosurgery, Yantai Yuhuangding Hospital, Yantai 264000, China; 7Department of Neurology, Tianjin Medical University General Hospital, Tianjin 300000, China; 8Department of Neurology, Yantai Yuhuangding Hospital, Yantai 264000, China; 9Department of Neurosurgery, First Hospital of Qinhuangdao, Qinhuangdao 066000, China

**Keywords:** TBI, abrocitinib, neuroinflammation, JAK1/STAT1/NF-κB

## Abstract

Background and Purpose: Neuroinflammation has been shown to play a critical role in secondary craniocerebral injury, leading to poor outcomes for TBI patients. Abrocitinib, a Janus kinase1 (JAK1) selective inhibitor approved to treat atopic dermatitis (AD) by the Food and Drug Administration (FDA), possesses a novel anti-inflammatory effect. In this study, we investigated whether abrocitinib could ameliorate neuroinflammation and exert a neuroprotective effect in traumatic brain injury (TBI) models. Methods: First, next-generation sequencing (NGS) was used to select genes closely related to neuroinflammation after TBI. Then, magnetic resonance imaging (MRI) was used to dynamically observe the changes in traumatic focus on the 1st, 3rd, and 7th days after the induction of fluid percussion injury (FPI). Moreover, abrocitinib’s effects on neurobehaviors were evaluated. A routine peripheral blood test was carried out and Evans blue dye extravasation, cerebral cortical blood flow, the levels of inflammatory cytokines, and changes in the numbers of inflammatory cells were evaluated to investigate the function of abrocitinib on the 1st day post-injury. Furthermore, the JAK1/signal transducer and activator of transcription1 (STAT1)/nuclear factor kappa (NF-κB) pathway was assessed. Results: In vivo, abrocitinib treatment was found to shrink the trauma lesions. Compared to the TBI group, the abrocitinib treatment group showed better neurological function, less blood-brain barrier (BBB) leakage, improved intracranial blood flow, relieved inflammatory cell infiltration, and reduced levels of inflammatory cytokines. In vitro, abrocitinib treatment was shown to reduce the pro-inflammatory M1 microglia phenotype and shift microglial polarization toward the anti-inflammatory M2 phenotype. The WB and IHC results showed that abrocitinib played a neuroprotective role by restraining JAK1/STAT1/NF-κB levels after TBI. Conclusions: Collectively, abrocitinib treatment after TBI is accompanied by improvements in neurological function consistent with radiological, histopathological, and biochemical changes. Therefore, abrocitinib can indeed reduce excessive neuroinflammation by restraining the JAK1/STAT1/NF-κB pathway.

## 1. Introduction

Traumatic brain injury (TBI), which is defined as cerebral dysfunction caused by external attacks, is one of the leading causes of death and disability worldwide [1,2]. The pathology of TBI can be categorized as primary brain injury which is caused by mechanical damage during the initial impact, or secondary brain injury, caused by inflammation, oxidative stress, mitochondrial dysfunction, apoptosis, etc. [3,4,5]. Among these pathological events, neuroinflammation, which is the main pathological process, can lead to extensive tissue injury and neurological dysfunctions [6,7]. As a result, secondary brain injury is evidenced by the secretion of pro-inflammatory cytokines, elevated blood-brain barrier (BBB) permeability, and the activation of the innate immune response in the central nervous system [8,9,10,11].

The Janus kinase/signal transducer and activator of transcription (JAK/STAT) signaling pathway is regarded as one of the most important cellular communication nodes. The JAK/STAT pathway regulates a variety of downstream events, including hematopoiesis, immune fitness, tissue repair, inflammation, apoptosis, and adipogenesis [12]. JAKl, JAK2, JAK3, and non-receptor tyrosine-protein kinase 2 (TYK2) are members of the JAK family, whereas STAT1, STAT2, STAT3, STAT4, STAT5a, STAT5b, and STAT6 are members of the STAT family. Cytokines, which combine with receptors, can subsequently induce the dimerization, phosphorylation, and activation of JAK. Activated JAK further induce the phosphorylation of members of the STAT family, which then form homo or heterodimers, transfer to the nucleus, and bind to specific DNA consensus sequences to dominate gene transcription [13,14]. JAK1, which can be phosphorylated by interleukin-(IL-)6, IL-2, interferon (IFN)α/β, IFN-γ, etc., is widely expressed in tissues and can phosphorylate all STATs [15,16]. STAT1 can be activated by IFN, IL-2, IL-6, and tumor necrosis factor (TNF) [17]. Then, STAT1 initiates the transcription of M1-associated genes, which can further lead to the production of inflammatory cytokines [18]. This promotes the expression of downstream inflammatory cytokines, resulting in an inflammatory reaction in hemorrhagic and ischemic brain tissues. In addition, STAT1 might contribute to brain damage by regulating the transcription and phosphorylation of proteins related to apoptosis [19].

According to the literature, the nuclear factor kappa B (NF-κB) signaling transduction pathway is directly involved in inflammation and apoptosis in TBI models [20]. After the stimulation of a TBI, the inhibitor of NF-κB (IκB) is phosphorylated and degraded, which leads to the release of the NF-κB dimer. Then, the activated NF-κB translocates to the nucleus, resulting in the transcription and activation of pro-inflammatory factors, such as NLR family pyrin domain containing protein 3 (NLRP3) inflammasomes, which are key elements in the procession of pyroptosis and are composed of an inflammasome sensor molecule, an apoptosis-associated speck-like protein containing a CARD (ASC), and caspase-1 [5,21].

Abrocitinib, a JAK1 selective inhibitor, was approved to treat atopic dermatitis (AD) by the FDA. Abrocitinib can effectively reduce autoimmune inflammation by blocking the JAK1’s ATP binding site and reversibly inhibiting the kinase. Given that neuroinflammation plays a vital role in the pathological development of TBI, we investigated whether abrocitinib could play an effective anti-inflammatory role in the early stages after TBI.

## 2. Materials and Methods

### 2.1. Animals

HFK Bioscience Corporation (Beijing, China) supplied 8-week-old male C57BL/6 mice. The mice were housed in a specific pathogen-free (SPF) environment and all animal experimental procedures were approved by the Animal Care and Use Committee of Tianjin Medical University General Hospital, Tianjin, China, and were carried out in accordance with institutional animal care guidelines.

The mice were randomly divided into three groups: Sham group, TBI + vehicle (saline) group, and TBI + abrocitinib group (10 mg/kg).

### 2.2. Fluid Percussion Injury (FPI) Model

The mice were anesthetized with isoflurane. The FPI model was prepared as referenced in previous research [22]. Briefly, a 3-mm-diameter cavity with intact dura matter (2.0-mm posterior from the bregma and 2.0-mm lateral to the sagittal suture) was used to rapidly inject saline at a regulated pressure into an unrestrained mouse utilizing an FPI device (Custom Design & Fabrication, Richmond, VA, USA), the pressure of percussion was manipulated at 1.9 ± 0.2 atm. The sham group received the same process except for the strike procedure. The sham group and TBI group were administered saline via the intragastric route, and the treatment group was administered abrocitinib via the intragastric route after TBI.

### 2.3. Bioinformatic Analysis

To identify the pathophysiological contributions in mouse brain tissue after TBI, we proceeded with transcriptome profiling in the TBI-treated brain tissues and those of the control group. The RNAseq datasets of the samples were divided into two groups correspondingly: The “control” group included five samples and the “TBI” group included five samples. Then, the differential expression of mRNA was investigated using the R package “DESeq2”. The threshold for mRNA differential expression screening was defined as “adjusted *p*-value < 0.05 and |log2 (fold change)| > 1”, and heatmaps and volcano plots were drawn accordingly. Using the R package “cluster profile”, the TBI-associated genes were screened, following that, gene ontology (GO) enrichment analysis (BP: Biological process; CC: Cellular component; MF: Molecular function) and Kyoto Encyclopedia of Genes and Genomes (KEGG) pathway analysis were carried out. To enrich the functional categories and pathways, a cut-off value of *p* < 0.05 was used. To determine the significant functional difference between the two groups, a gene set enrichment analysis (GSEA) was performed using the R package “cluster profile”. Herein, we used the *p*-value and normalized enrichment score (NES) to organize the enriched pathways in each cluster. |NES| > 1 and *p* < 0.05 were considered to be significantly enriched gene sets. Furthermore, the analysis used CIBERSORT to assess the immune response of 25 types of immune cells in TBI to determine their correlation with survival and molecular subpopulation. Finally, we compared the data from the two groups and then used the data to generate a heatmap and a violin plot.

### 2.4. MRI

All MRI images were acquired using a 9.4 T Bruker BioSpec 94/30 USR MRI scanner (Bruker BioSpin GmbH, Ettlingen, Germany). On the 1st, 3rd, and 7th days after FPI induction, the animals were transferred to an MRI-compatible animal bed in the prone position and a mouth and nose mask was applied to deliver 1.5% isoflurane in oxygen at a delivery rate of 500 mL/min. A volume coil with an inner diameter of 86 mm was used for RF transmission, and a mouse brain surface coil was used to receive signals. After a localizer image was scanned to ensure the animal’s head was well positioned, a coronal T2WI was acquired with the following parameters: Repetition time = 2500 ms; echo time = 33 ms; number of averages = 2; slice thickness = 0.5 mm with 20 slices; field of view = 20 × 20 mm; matrix size = 256 × 256, yielding an imaging resolution of 0.078 × 0.078 mm; scanning time = 160 s. Sante Dicom Viewer and Image J were used to calculate the lesion volume. 

### 2.5. Hematoxylin and Eosin (H&E) Staining and Nissl Staining

Anesthetized mice were perfused with PBS and paraformaldehyde on the 1st day after FPI. Brain tissues were collected, embedded in paraffin blocks, and cut into 5 μm slices. After dewaxing, according to the manufacturer’s instructions, hematoxylin and eosin staining kit (CAT#G1120, Solarbio, Beijing, China) was used to stain the paraffin sections. For Nissl staining, after dewaxing, the slices were soaked in three times distilled water for 5 min and stained with methyl violet staining solution (BP034, Biossci biotechnology Co.; Ltd., Wuhan, China) in a 56 °C incubator for 20 min. The sections were washed with distilled water and divided with 70% ethanol for 1 min. Then, 0.1% glacial acetic acid was used to distinguish the slides, and 100% ethanol solution and xylene were used to successively dehydrate and clear the slides, respectively. The slices were mounted with neutral glue and observed under a light microscope (BX46, Olympus Corporation, Tokyo, Japan) for the assessment of neuron numbers. 

### 2.6. Evans Blue Dye Extravasation

To observe the FPI-induced vascular leakage and brain edema, through the tail vein, mice received 100 µL of 2% Evans blue (Sigma Aldrich, St. Louis, MO, USA). Two hours after injection, the mice were sacrificed and given PBS perfusions. After dissecting the brain, the horizontal and coronal planes of the tissues were observed under a stereoscope. After the removal of the cerebellum and olfactory bulb, the half-brains were weighed and homogenized in 1 mL formamide (Aladdin, Shanghai, China) before incubation at 60 °C overnight. For 30 min, the brain homogenates were centrifuged at 14,000 rpm. The Evans blue in the supernatant was transferred to a 96-well plate with 200 µL per well and set in a duplicate hole; then, the plate was measured at OD620 nm in a preheated microplate reader. A standard curve was defined using Evans blue solution with doubling dilution configurations (50, 25, 12.5, 6.25, 3.12, 1.56, 0.78, and 0 µg/mL), and the content of Evans blue (µg/mL) in the samples was calculated. Then, the content of Evans blue per gram of brain tissue (µg/g) was calculated as the content of Evans blue in the sample (µg/mL) × formamide volume (mL)/brain weight (g).

### 2.7. Cerebral Blood Flow

Cerebral cortical blood flow was monitored for 24 h post-TBI using a non-invasive laser speckle imager with a 70 mW built-in laser and a 1388 × 1038-pixel CCD camera (PeriCam PSI System, Perimed, Sweden) as recently described [23]. Briefly, mice were sedated with isoflurane and placed on a temperature-controlled stereotaxic frame (RWD Life Science, San Diego, CA, USA), with the skull exposed. The CCD camera was installed 10 cm above the skull to capture the CBF images in a scanning area of 2.0 × 2.0 cm. The ipsilateral cortical blood flow (perfusion unit, PU) was calculated and presented as mean perfusion values using the vendor-supplied PIMSoft software (version 1.5; Perimed, Jakobsberg, Sweden).

### 2.8. Modified Neurological Behavioral Tests

Prior to the induction of FPI, a modified neurological severity scores (mNSS) functional test was performed, and it was repeated on the 1st, 3rd, 5th, and 7th days after TBI. The procedures used were based on previous research [24]. All testers were well-trained and blinded to the experimental treatment assignment.

### 2.9. IHC Staining

Anesthetized mice were sacrificed on the 1st day after injury and their brain tissues were embedded in paraffin blocks and cut into 5-μm slices according to the standard procedures. After a series of dewaxing and hydration, the slides were boiled for 30 min in sodium citrate buffer (0.01 mol/L, pH 6.0) (C1010, Solarbio, Beijing, China) for antigen retrieval. Then, the slides were blocked using PBS, which contains 0.3% Triton X-100 (#T8200 Solarbio, China) and 3% albumin bovine V (A6020 Biotopped, Beijing, China) for 1.5 h at room temperature to inhibit nonspecific binding before the sections were incubated overnight at 4 °C with primary antibodies against Iba-1 (1:800 #17198 Cell signaling, Danvers, MA, USA), MPO (1:200 ab65871 Abcam, Cambridge, UK), and GSDMD (1:200 ab219800 Abcam, Cambridge, UK). The slides were washed with PBS and then incubated with a secondary antibody (GK500705 Gene Tech, Shanghai, China) for 1 h. After washing with PBS, 3,3-diaminobenzidine (1:500 DAB GK500705 Gene Tech, Shanghai, China) was added for the horseradish peroxidase reaction. Then, the sections were re-stained with hematoxylin (#g1120 Solarbio, Beijing, China), dehydrated, and cleared using gradient ethanol and xylene. Finally, the slides were observed under an *u-lhledc* microscope (BX46, Olympus Corporation, Tokyo, Japan) after being sealed with neutral glue. The ImageJ software (version 1.53, National Institutes of Health, Bethesda, MD, USA) was used to analyze the IHC images.

### 2.10. TUNEL Staining

As previously described [25], frozen 6-μm-thick tissue sections were prepared. After 24 h in 4% paraformaldehyde, the brain tissues were dehydrated in 15% sucrose until they sank to the bottom. Then, the tissues were dehydrated in 30% sucrose until they sank to the bottom. After 10 min of immersion in the OCT compound (4583 SAKURA, Torrance, CA, USA), the tissues were frozen with liquid nitrogen. Thereafter, the brain tissues were embedded in frozen blocks cut into 6-μm sections. Bovine serum albumin (3%) in phosphate-buffered saline (PBS) with 0.3% Triton X-100 was used to block the sections. TUNEL staining was performed by incubating the sections in TUNEL (C1088 Beyotime, Shanghai, China) for 1 h at room temperature. After three washes with PBS, a mounting medium containing 4′,6-diamidino-2-phenylindole (DAPI) was applied to the sections. Then, the slices were sealed with a coverslip. Finally, fluorescent signals were discovered using a fluorescence microscope (Olympus BX61, Tokyo, Japan). The IF images were analyzed using ImageJ software (version 1.53, National Institutes of Health, USA).

### 2.11. Cell Culture, LPS Model, Experimental Design, and Drug Administration

DMEM/F-12 medium (Gibco, New York, USA) supplemented with 5% fetal bovine serum (Sigma-Aldrich, St. Louis, MO, USA) and 1% penicillin-streptomycin (Hyclone, Logan, UT, USA) was used to culture mouse BV2 microglial cells (American Type Culture Collection, Manassas, VA, USA). The cells were incubated at 37 °C in a humidified 5% carbon dioxide atmosphere with the medium changed every 2 days. The cells were exposed to LPS to mimic cerebral neuroinflammation after TBI in vitro. To evaluate its effects on microglial polarization, abrocitinib (100 nM, 500 nM, or 1 µΜ; Selleck, Houston, TX, USA) was added to the culture medium for the management of LPS (10 µg/mL; Tocris Bioscience, Bristol, UK). The treated BV2 cells were collected after 6 h of LPS irritation for further immunofluorescence (IF) staining.

### 2.12. Immunofluorescence Staining

For IF staining, a sterile round glass slide (18 mm) was placed on a 12-well plate and coated with 1× poly-lysine overnight. BV2 cells were seeded into the 12-well plate at a concentration of 1 × 10^5^/cells the next day. Following the intervention, the treated cells were fixed for 20 min in precooled 4% PFA and permeabilized for 20 min at room temperature with 0.1% Triton X-100 (#T8200 Solarbio, Beijing, China). The fixed cells were blocked for 30 min at 37 °C with 3% bovine serum albumin (A6020 Biotopped, Beijing, China) and the sections were incubated overnight at 4 °C with primary antibodies against Iba-1 (1:800 #17198 Cell signaling, Danvers, MA, USA), iNOS (1:500 ab210823, Abcam, Cambridge, UK), and Arg-1(1:200 ab239731, Abcam, Cambridge, UK). Then, the sections were rinsed three times with PBS for 5 min before incubation for 1 h at room temperature with Alexa Fluor 488 goat anti-rabbit IgG (1:500, a-11008, Thermo Fisher Science, Waltham, MA, USA) and Alexa Fluor 555 donkey anti-mouse IgG (1:500, a-31570, Thermo Fisher science, MA, USA). Prior to the addition of DAPI, the sections were washed three times with PBS and then sealed with a coverslip. A fluorescence microscope (Olympus BX61, Tokyo, Japan) was used to observe the signals. The IF images were analyzed using ImageJ software (version 1.53, National Institutes of Health, USA).

### 2.13. Routine Analysis of Blood

Blood was collected from the apex of the heart in 400 µL aliquots (*n* = 8/group) on the 1st day after injury. The blood was stored in anticoagulant tubes containing EDTAK2. Then, the samples were sent to a clinical laboratory where they were counted using an automatic blood analyzer (BC-2800Vet MINDRAY, Shenzhen, China).

### 2.14. Enzyme-Linked Immunosorbent Assay (ELISA)

The brain samples were homogenized according to the manufacturer’s protocol. Inflammatory factors were detected in the brain samples using ELISA kits for TNF-α, IL-1β, IL-6, and IL-18 (RK00027, RK00006, RK00008, RK00104, Abclonal, Wuhan, China). The measured OD values were converted into concentration values.

### 2.15. Western Blot

WB was carried out in the manner previously described [26]. Total proteins from each group’s brain tissues were extracted using lysis buffer containing protease and phosphatase inhibitors (P1260 Solarbio, Beijing, China). The tissue was ground up, and the protein was then gathered in the supernatant and measured with the bicinchoninic acid technique. Specific primary antibodies were incubated with the PVDF membranes overnight at 4 °C. The primary antibodies used were as follows: Primary rabbit monoclonal against JAK1 (1:1000, A11963; Abclonal, Wuhan, China); phospho-JAK1 (1:1000, AP0530; Abclonal, Wuhan, China); STAT1 (1:1000, A19563; Abclonal, Wuhan, China); phospho-STAT1 (1:1000, AP0054; Abclonal, Wuhan, China); phospho-NF-κB p65 (1:1000, #3033; Cell Signaling, Danvers, MA, USA); ASC (1:1000, NBP1-78977; NOVUS, Colorado, USA); caspase-1 (1:1000, A0964; Abclonal, Wuhan, China); GSDMD (1:1000 A20728, Abclonal, Wuhan, China); NRLP3 (1:1000, A5652; Abclonal, Wuhan, China); and primary mouse polyclonal anti-GAPDH (1:1000, #97166; Cell Signaling, Danvers, MA, USA). The membranes were washed three times after incubation. Then, the membranes were incubated at room temperature for 1 h with goat anti-rabbit (1:5000; Zhongshan Golden Bridge, Beijing, China) and goat anti-mouse IgG antibodies (1:5000; Zhongshan Golden Bridge, Beijing, China). Finally, the membranes were exposed to the ChemiDoc Touch Imaging System (Bio-Rad, Hercules, CA, USA) after incubation with enhanced chemiluminescence. ImageJ was used to perform the densitometric analysis on protein bands.

### 2.16. Statistical Analysis

All data were expressed as mean ± standard deviation (SD). An F-test was performed to evaluate the homogeneity of variance and the Kolmogorov-Smirnov test was used for quantitative data to determine whether the data were normally distributed. If the distribution was normal and the variance was homogeneous, a one-way or repeated measures analysis of variance (ANOVA) test was used among the groups, and Tukey’s test was used as a post-hoc test. Otherwise, a Kruskal-Wallis test was used among the groups. The Statistical Package for the Social Sciences (SPSS) software version 26.0 (IBM Corporation, Armonk, NY, USA) was used to conduct the statistical analysis. A *p*-value less than 0.05 was considered significant. 

## 3. Results

### 3.1. Sequence-to-Function Analysis via Next-Generation Sequencing (NGS)

We started by analyzing the genes that showed differential expression in mice after TBI and the sham operation using the transcriptome profiling database. According to the generated volcano plot (Figure 1A) and heatmap (Figure 1B), there were 436 differentially expressed genes in mice from the two groups, including 359 upregulated genes and 77 downregulated genes. The expressions of TNF-α, IL-1β, IL-6, NLRP3, and ASC were found to be significantly higher in the TBI group (Figure 1C). Following the analysis of differentially expressed genes in the aforementioned database, we performed a GO and KEGG analysis of the 359 upregulated and 77 downregulated genes. The results of the GO analysis (Figure 1D) showed that TBI has important roles in cytokine activity, the inflammasome complex, the acute inflammatory response, and neutrophil migration. The results of the KEGG analysis showed that TBI has important roles in cytokine-cytokine receptor interaction, the NF-κB signaling pathway, the JAK-STAT signaling pathway, and the NOD-like receptor signaling pathway (Figure 1E). Additionally, the GSEA results showed that the GO items apoptotic process, cytokine activity, inflammatory response, innate immune response, and neutrophil migration showed significantly differential enrichment in the TBI group based on the NES and p-value (Figure 1F), while the KEGG items apoptosis, cytokine-cytokine receptor interaction, NF-κB signaling pathway, JAK-STAT signaling pathway, and NOD-like receptor signaling pathway were shown to be significantly differentially enriched in the TBI group based on the NES and *p*-value (Figure 1G). These results indicate that the acute inflammatory response plays a major role in the early stage after TBI in mice. We used an established computational resource (CIBERSORT) to explore the gene expression profiles of our dataset to infer the density of 25 types of immune cells. Figure 1H shows the proportions of the 25 species of subpopulations of immune cells in each sample from the two groups. Neutrophil cells and M1 macrophages were the main immune cells affected by TBI, as indicated by an increase in the TBI group compared to the control group (Figure 1I).

### 3.2. Abrocitinib Ameliorated Brain Injury and Improved Neurological Outcomes after TBI 

Through dynamic MRI observation on the 1st, 3rd, and 7th day after TBI, we found that the size of the trauma focus was gradually reduced in the first 7 days after TBI. Compared to the TBI group, the abrocitinib treatment significantly controlled the size of the trauma focus and enhanced the absorption of the hematoma (Figure 2A,B). The Evans blue dye extravasation test was performed on the 1st day after TBI. In the acute phase after TBI, the damage to the blood-brain barrier caused by TBI was significantly ameliorated by abrocitinib (Figure 2D,E). Moreover, drawing support from the results obtained using the non-invasive laser speckle imager, abrocitinib treatment significantly increased cortical cerebral blood flow (CBF) when compared to mice receiving saline (Figure 2F,G). On the one hand, all mice with TBI had severe unilateral sensorimotor deficits. On the other hand, a tendency for the mNSS to decrease over time after TBI was revealed. However, abrocitinib induced a significant decrease in the mNSS on the 3rd day after treatment when compared to the animals treated with saline, which indicated that the neurological function was apparently improved by the abrocitinib treatment (Figure 2C). 

### 3.3. Abrocitinib Promoted the Survival of Neurons and Reduced Apoptosis

First, from the H&E staining, there was an intuitive sense of the degree of damage to the cortex. The H&E staining indicated that the abrocitinib treatment shrunk the trauma focus, which was in agreement with the MRI results discussed above (Figure 3A). Nissl staining was employed to determine the number of neurons in the peripheral areas of the trauma focus. In both experimental groups where TBI was induced, the number of neurons decreased significantly. However, the number of neurons in the abrocitinib treatment group was significantly higher than the TBI group (Figure 3B,D). Moreover, TUNEL fluorescent staining was performed to measure the change in apoptosis around the trauma focus. We found that TBI induced significant apoptosis around the wound focus. Abrocitinib reduced the number of apoptotic cells on the 1st day after TBI (Figure 3C,E). Therefore, it was clear that abrocitinib played an anti-apoptotic and neuroprotective role. 

### 3.4. Abrocitinib Significantly Inhibited Activation of JAK1/STAT1

A Western blot was used to determine the expression of JAK1/STAT1 and p-JAK1/p-STAT1. The results showed that after TBI, there was a significant decrease in the total content of JAK1 and STAT1. After abrocitinib treatment, the expressions of p-JAK1 and p-STAT1 increased dramatically in the TBI group. In addition, this phenomenon could be remarkably restrained by abrocitinib. Based on the expressions of p-JAK1 and p-STAT1, JAK1 and STAT1 were significantly activated on the 1st day after TBI. After abrocitinib treatment, the levels of p-JAK1 and p-STAT1 declined significantly, which meant that abrocitinib effectively depressed the activation of JAK1 and STAT1 (Figure 4). 

### 3.5. Abrocitinib Did Not Change Blood Counts after TBI

To observe whether abrocitinib had effects on the hematopoietic system, peripheral blood counts were measured. As shown in Table 1, the numbers of white blood cells, red blood cells, platelets, and hemoglobin did not change evidently among the three groups. However, compared to the sham group, the number of neutrophils and the percentage of neutrophils were significantly increased in the TBI and TBI + abrocitinib groups. 

### 3.6. Abrocitinib Reduced the Infiltration of Inflammatory Cells and the Activation of Microglia, Inhibited M1 Polarization and Promoted M2 Polarization, and Further Decreased Pro-Inflammatory Cytokines

To observe the infiltration of inflammatory cells, IHC staining was performed to distinguish neutrophils and microglia (Figure 5A–D). The results of the IHC staining indicated that the number of neutrophils and microglia increased significantly in the acute phase after TBI. After abrocitinib intervention, the numbers of both types of inflammatory cells decreased significantly. Moreover, using surface markers specific for M1 and M2 in vitro, the state of microglial/macrophage polarization was detected. After abrocitinib treatment, the number of activated microglia/macrophages (Iba-1^+^) that expressed the M1 marker iNOS after LPS irritation markedly reduced, while the expressed M2 marker Arg-1 increased, indicating an M1-to-M2 transition. Moreover, this phenomenon became more apparent with the increasing dosage of abrocitinib (Figure 5E). TBI caused elevated pro-inflammatory factors in the brain tissue. We measured the levels of IL-6 and TNF-α in the brain tissues to investigate the role of the inflammatory response after TBI. When compared to the sham group, the ELISA results showed that TBI induced the secretion of these pro-inflammatory cytokines. On the other hand, abrocitinib could reverse the increased levels of IL-6 and TNF-α after TBI (Figure 5F,G).

### 3.7. Abrocitinib Induced Significant Anti-Inflammatory Effects and Decreased Pyroptosis in Brain Tissue after TBI by Restraining the NF-κB Pathway

NF-κB is still recognized as a predominant transcription factor in the regulation of pro-inflammatory mediators after TBI [27]. TBI increases the levels of pro-inflammatory factors in brain tissue, some of which, such as IL-1β and IL-18, are important in the process of pyroptosis. Moreover, the NF-κB-related signaling pathway was determined to investigate the role of the inflammatory response and pyroptosis after TBI. The WB results revealed that following TBI, the content of p-NF-κB evidently increased, as did the levels of NLRP3, ASC, caspase-1, and GSDMD, which are the downstream factors of NF-κB (Figure 6A). In addition, the contents of IL-1β and IL-18 were drastically increased after TBI (Figure 6D,E). All factors not only led to an excessive inflammatory response, but also exacerbated pyroptosis, which was investigated using GSDMD IHC staining (Figure 6B,C). Conversely, abrocitinib treatment significantly reduced the levels of NF-κB and its downstream factors. Therefore, the pyroptosis induced by those factors lessened, which was verified by WB results and GSDMD IHC staining (Figure 6). 

## 4. Discussion

Due to cell death and neuronal dysfunction in the surrounding and remote brain areas, secondary craniocerebral trauma, which includes calcium excess, glutamatergic excitotoxicity, oxidative stress, and the inflammatory response, can cause functional loss and have a negative prognosis [3,28,29,30]. Recently, increasing evidence from human and animal studies has indicated that persistent and excessive secondary injuries are averse to the recovery of neural function after TBI [31,32]. In addition, excessive neuroinflammation which is capable of initiating neuronal cell death and BBB disruption, is considered to be an important molecular and cellular characteristic of TBI [6,9,33]. Under normal circumstances, microglia remain in a quiescent state and perform surveillance functions in the CNS. However, under pathological conditions, microglial cells are rapidly activated, leading to the release of cytokines, chemokines, reactive oxygen/nitrogen species (ROS/RNS), and prostaglandins. Furthermore, cytokines, chemokines, ROS, and several proteases discharged by activated microglia can aggravate their neurotoxic effects [34,35,36]. In summary, a sterile immune response that comprises local signaling in neurons, glia, and recruited peripheral immune cells is established and triggers an inflammatory cascade within minutes of TBI [37]. Although balanced inflammation is required for cellular debris removal and tissue remodeling, persistent and excessive inflammation can worsen neuronal apoptosis and neurological damage following TBI [38,39]. As a result, effectively controlling excessive neuroinflammation promotes neuroprotection and neural function recovery, potentially improving TBI prognosis.

In our experiments, the NGS results reflected that inflammation-related pathways, including JAK/STAT, NF-κB, etc., were activated in the early stage of TBI. By analyzing inflammatory cells, we found that microglia and neutrophils were significantly activated in the early period post-TBI. In addition, these results, which were generated using NGS, were verified in subsequent experiments. It has been noted in the literature that in the initial stages following TBI, microglia are activated and release proinflammatory cytokines, such as TNF-α, IL-1β, IL-18, and IL-6, which results in a local neuroinflammatory response. Furthermore, these inflammatory mediators may mobilize immune cells to the injury regions, resulting in additional neuroinflammatory reactions [40]. In our experiments, microglia content was dramatically increased, as were the levels of proinflammatory cytokines including TNF-α, IL-1β, IL-18, and IL-6. Subsequently, neutrophils were recruited to the periphery of the wound focus, leading to a severe inflammatory reaction. Moreover, JAKs can be activated by the extracellular interaction of cytokines with the appropriate membrane receptors, as having two JAK proteins in adjacent proximity permits phosphorylation to take place [41]. Some cytokines, for example, IL-6, possess the ability to activate JAK1. P-JAK1 is a significant upstream protein for STAT1. After phosphorylation by p-JAK1, STAT1 dimerizes and transmigrates to the nucleus and then transactivates genes that respond to STAT1 [14,42]. Following this, P-STAT1 stimulates the transcription of M1-associated genes, which ultimately leads to the creation of inflammatory cytokines [43]. IL-1β, IL-6, TNF-α, and IL-18 are cytokines that are generated when microglia are activated, and induce neuroinflammation and neurodegeneration [44]. Therefore, the inflammatory response is greatly influenced by the JAK1/STAT1 signaling pathway [45]. In the process of secondary brain injury, a positive feedback loop develops that may cause neuroinflammation to be more pronounced. If some factors in this loop could be suppressed, the neuroinflammation could possibly be attenuated to some extent. Here, we used abrocitinib, a JAK1 inhibitor, to achieve the goal of breaking this vicious cycle and reducing excessive neuroinflammation. From the results of our experiments, we found that after abrocitinib treatment, there was no effect on the hematopoietic system. Furthermore, brain edema was effectively controlled, the speed of hematoma absorption was enhanced, local blood flow around the wound focus improved, the numbers of microglia and neutrophils and contents of inflammatory cytokines decreased, neuronal apoptosis decreased, and neurological recovery was facilitated. Moreover, in our in vitro experiments, abrocitinib inhibited the activation of M1 microglia and promoted their transformation to M2 microglia, resulting in an attenuated inflammatory response. In summary, abrocitinib effectively restrained the activation of JAK1. As a result, the contents of p-JAK1 and p-STAT1 decreased significantly, the levels of activated microglia decreased and the M1-to-M2 transition increased, and the subsequent infiltration of neutrophils and production of inflammatory factors lessened. Finally, the excessive inflammatory reaction sequentially decreased.

Previous studies have shown that inflammatory signaling pathways, such as NF-κB and JAKs/STATs, can interact with one another and play critical roles in triggering, promoting, and regulating inflammatory responses [46]. NF-κB can induce STAT1 synthesis and promote STAT1 activation. In fact, multiple studies have demonstrated that the regulation of IFN-γ-induced inflammatory genes involves interactions between NF-κB and STAT1 signaling pathways [47,48,49]. In this investigation, we also demonstrated that abrocitinib could significantly reduce the elevated NF-κB content caused by TBI. NF-κB is a key transcription factor that plays a role in inflammatory responses that occur after neuroinflammatory disorders [50]. NF-κB signaling could be activated by hypoxia, reactive oxygen species, and several inflammatory mediators [51]. Once activated, NF-κB can cause the production of proinflammatory cytokines, such as IL-1β, IL-6, and TNF-α, as well as the activation of NLRP3 inflammasome [5,52]. Additionally, some inflammatory cytokines, such as IL-1 and TNF-α, can activate NF-κB further, creating a positive feedback loop that exacerbates the inflammatory response [19,53]. Inflammasomes are multimeric protein complexes composed of an inflammasome sensor molecule, an apoptosis-associated speck-like protein containing a CARD (ASC) and caspase-1 [54]. Inflammasomes activate caspase-1 and cause the release of IL-1β, IL-18, and gasdermin D (GSDMD), which promotes pyroptosis [55,56]. Infections of the central nervous system, cell death, brain injury, and neurodegenerative disease all rely on IL-1β and IL-18. IL-1β and IL-18 concentrations are higher in various neurodegenerative disorders [57]. In our study, after treatment with abrocitinib, the content of NF-κB decreased, which resulted in a reduction in the levels of NLRP3 and other inflammatory cytokines. In addition, the decrease in NLRP3 resulted in low expressions of ASC, caspase-1, and GSDMD, which suggested that the pyroptosis process after TBI was significantly inhibited. The high survival rate of neurons and decrease in apoptotic cells would be beneficial to the recovery of nerve function, which means the prognosis of TBI can be greatly improved. Indeed, our research also has some unresolved issues. One day after TBI we established the observation window according to the NGS results. If we increased the dosage and prolonged the time of administration of abrocitinib, what impacts would it have on long-term rehabilitation after TBI? These problems are urgent and should be solved in future research.

In summary, based on FPI models, we confirmed that in the early period post-TBI, abrocitinib not only modulated the release of inflammatory cytokines, but also promoted the shrinkage of brain edema and hematoma and improved neural functions in the mice with TBI. The therapeutic results of abrocitinib treatment might be partially ascribed to the normalization of the inflammatory reaction and the effects of restraining JAK1/STAT1/NF-κB pathways. Abrocitinib may be a promising candidate for the treatment of secondary brain injury. Further rigorous clinical evaluations must still be conducted.

## Figures and Tables

**Figure 1 cells-11-03588-f001:**
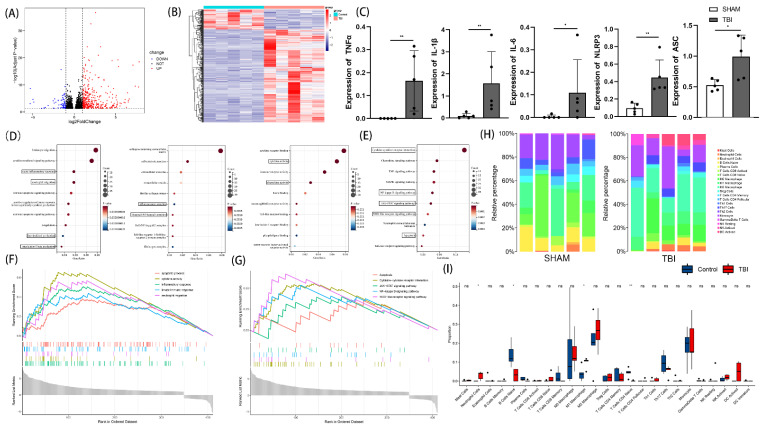
Gene expression analysis for TBI and control mice. Volcano plot of differential analysis (**A**). Heatmap of differential analysis (**B**). Comparison of TNF-α, IL-1β, IL-6, NLRP3, and ASC expression between TBI group and control group (**C**) (n = 5, Kruskal-Wallis test). Bubble chart of BP, CC, and MF analysis (**D**). Bubble chart of KEGG analysis (**E**). Enrichment plots for GO and KEGG items from gene set enrichment analysis (GSEA) (**F**,**G**). Heatmap of immune infiltration in sham and TBI groups (**H**). Violin plot of immune infiltration in sham and TBI groups (**I**). All data are shown as mean ± SD. * *p* < 0.05, ** *p* < 0.01.

**Figure 2 cells-11-03588-f002:**
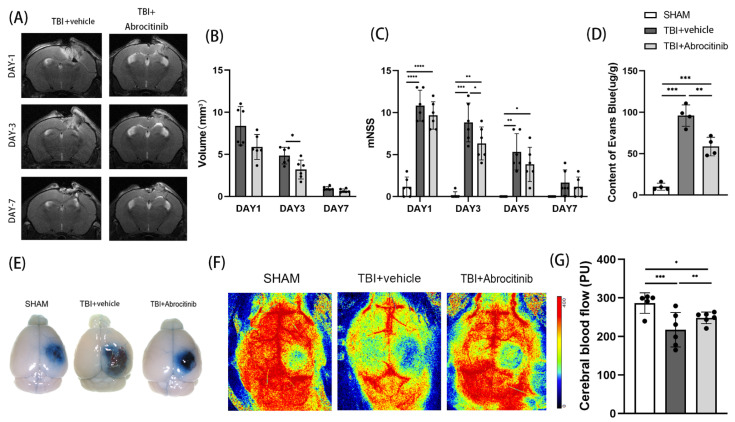
Effects of abrocitinib on hematoma absorption, the results of behavioral tests, cerebral edema, and cerebral cortical blood flow. Changes in trauma focus were dynamically observed with the help of MRI (**A**,**B**) (n = 6, repeated measures ANOVA with Tukey’s post-hoc test). In addition, neural function was evaluated using mNSS scores (**C**) (n = 6, repeated measures ANOVA with Tukey’s post-hoc test). FPI-induced vascular leakage was measured via Evans blue dye extravasation (**D**,**E**) (n = 4, Kruskal-Wallis test) and cerebral cortical blood flow (**F**,**G**) (n = 5–6, one-way ANOVA with Tukey’s post-hoc test) was also monitored on the 1st day after TBI. All data are shown as mean ± SD. * *p* < 0.05, ** *p* < 0.01, *** *p* < 0.001, **** *p* < 0.0001.

**Figure 3 cells-11-03588-f003:**
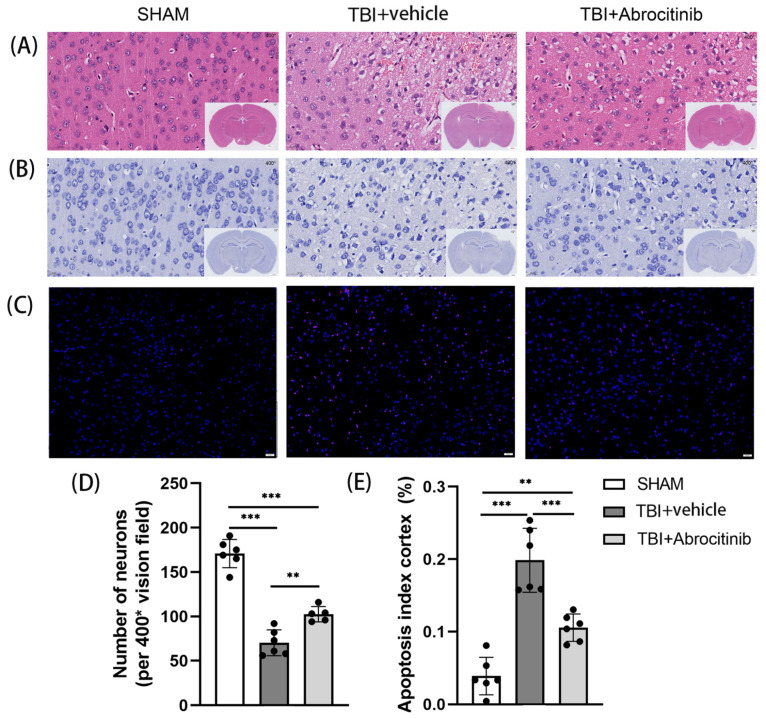
Changes in pathological morphology. Morphological changes were observed using H&E staining (**A**). The number of surviving neurons (**B**,**D**) and the changes in apoptotic cells around the wound focus were observed via Nissl staining and TUNEL staining, respectively (DAPI—blue; TUNEL—red) (**C**,**E**) (n = 6, one-way ANOVA with Tukey’s post-hoc test). All data are shown as mean ± SD. * *p* < 0.05, ** *p* < 0.01, *** *p* < 0.001.

**Figure 4 cells-11-03588-f004:**
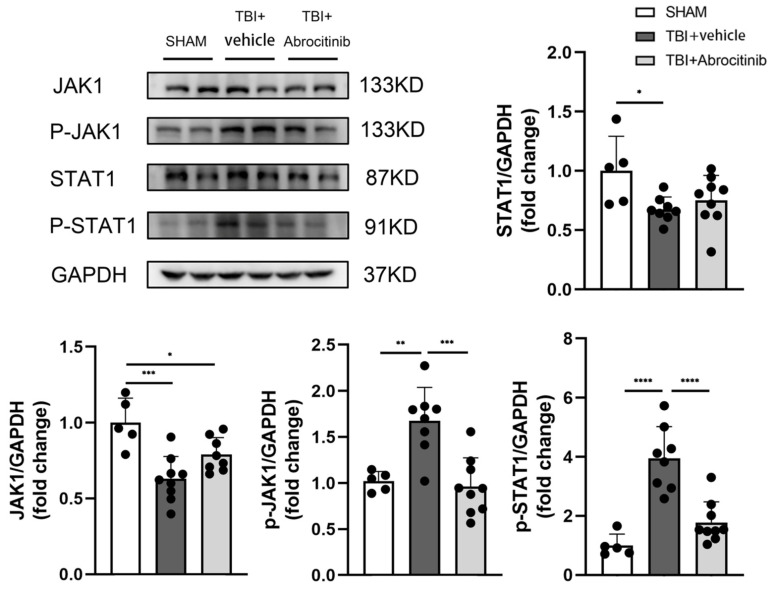
Effects of abrocitinib on the JAK1/STAT1 pathway. On the 1st day after TBI, JAK1 and STAT1 showed a dramatic increase in activation. After abrocitinib treatment, the activated JAK1 and STAT1 were significantly decreased, as shown in the WB results above and can be displayed through WB (n = 5–9, one-way ANOVA with Tukey’s post-hoc test). All data are shown as mean ± SD. * *p* < 0.05, ** *p* < 0.01, *** *p* < 0.001, **** *p* < 0.0001.

**Figure 5 cells-11-03588-f005:**
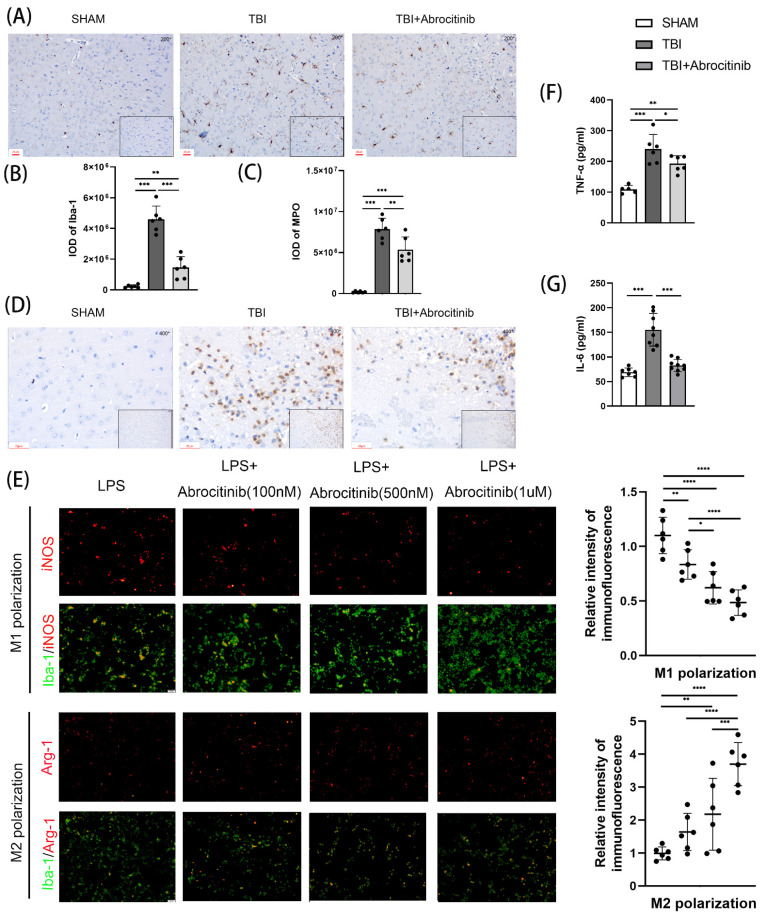
Changes in the infiltration of inflammatory cells and the activation of microglia. From the IHC staining, the increased neutrophils and microglia around the trauma focus caused by TBI were attenuated by abrocitinib (**A**–**D**) (n = 6 Kruskal-Wallis test). The effect of abrocitinib on LPS-mediated BV2 microglial polarization was also observed in vitro (**E**). M1-phenotype: iNOS + (red) and Iba-1 + (green); M2-phenotype: Arg-1 + (red) and Iba-1 + (green). Scale bar = 20 μm (n = 6 one-way ANOVA with Tukey’s post-hoc test). The changes in inflammatory cytokines (IL-6 and TNF-α) after brain injury and the effects of abrocitinib were determined with the help of ELISA (**F**–**G**) (n = 5–9 one-way ANOVA with Tukey’s post-hoc test). All data are shown as mean ± SD. * *p* < 0.05, ** *p* < 0.01, *** *p* < 0.001, **** *p* < 0.0001.

**Figure 6 cells-11-03588-f006:**
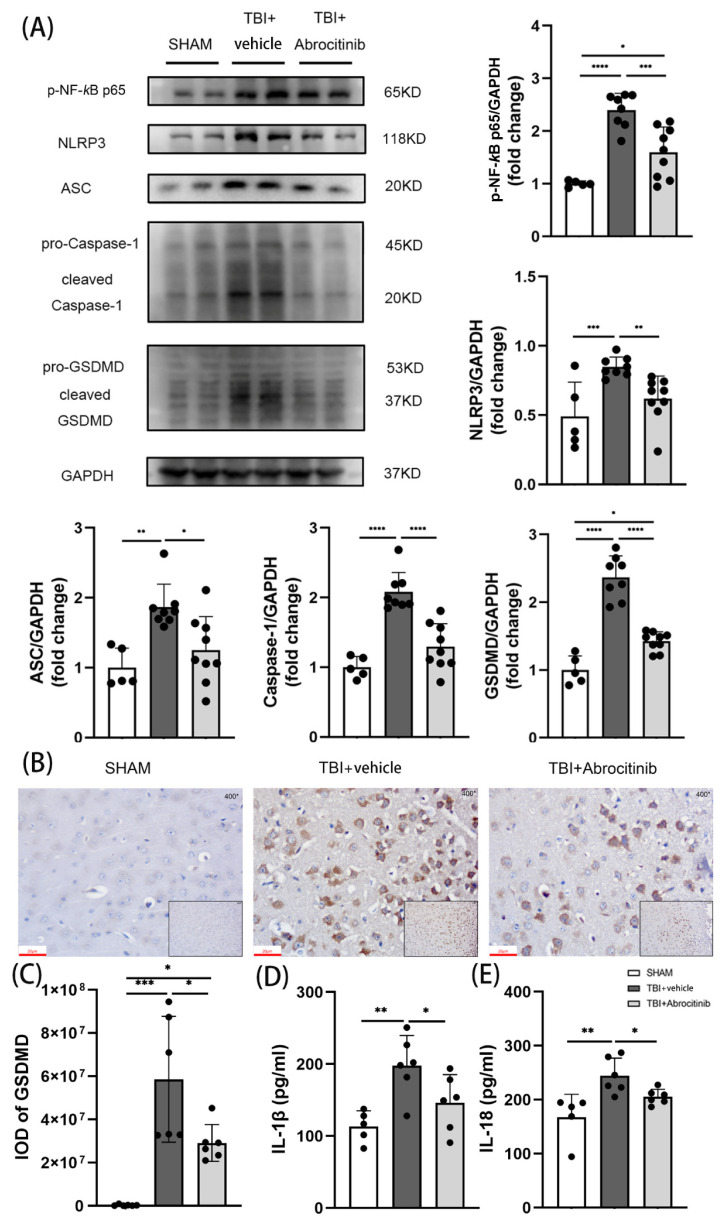
Effects of abrocitinib on NFκB-related inflammation and pyroptosis pathways. On the 1st day after TBI, NFκB-related inflammation and the activation of pyroptosis pathways were dramatically increased. After abrocitinib treatment, the indicators of NFκB-related inflammation and pyroptosis pathways were significantly decreased, as can be seen from the WB (n = 5–9 one-way ANOVA with Tukey’s post-hoc test) and GSDMD IHC staining (**A**–**C**) (n = 6 Kruskal-Wallis test). From the ELISA results, the changes in inflammatory cytokines (IL-1β and IL-18) after brain injury and the effects of abrocitinib were precisely revealed (**D**,**E**) (n = 5–6 one-way ANOVA with Tukey’s post-hoc test). All data are shown as mean ± SD. * *p* < 0.05, ** *p* < 0.01, *** *p* < 0.001, **** *p* < 0.0001.

**Table 1 cells-11-03588-t001:** Routine analysis of blood, except for changes in neutrophils, the blood counts of the TBI mice receiving abrocitinib or saline remained stable after treatment initiation (*p* > 0.05). Eight samples were included in each group (Kruskal-Wallis test). * *p* < 0.05, ** *p* < 0.01.

	Sham	TBI + Vehicle	TBI + Abrocitinib
Number of White Blood Cells (×10^9^/L)	2.41 ± 1.36	2.50 ± 1.28	2.81 ± 0.99
Number of Neutrophils (×10^9^/L)	0.33 ± 0.09	0.58 ± 0.24 *	0.57 ± 0.23 *
Number of Red Blood Cells (×10^12^/L)	8.27 ± 0.53	8.22 ± 0.97	8.28 ± 0.49
Number of Platelets (×10^9^/L)	659.14 ± 180.31	687.88 ± 118.02	716.71 ± 84.52
Percentage of Neutrophils (%)	12.77 ± 3.05	24.93 ± 9.44 **	20.45 ± 3.09 *
Hemoglobin (g/L)	121.0 ± 5.88	125.25 ± 6.16	118.86 ± 6.47

## Data Availability

All data generated or analyzed during this study are included in this published article.

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
