# Peer review of "Abrocitinib Attenuates Microglia-Mediated Neuroinflammation after Traumatic Brain Injury via Inhibiting the JAK1/STAT1/NF-κB Pathway"

_cells, 2022, doi:10.3390/cells11223588_

Round 1
Reviewer 1 Report
The authors evidenced that the abrocitinib treatment after TBI improves the neurological function consistent with radiological, histopathological, and biochemical changes. The characterization of microglia types and neuroinflammation were particular interested in this filed. In addition, the conclusions were supported by the well organized experiments. I have no further concerns in the present studies by the authors.
Author Response
Thank you for your recognition and suggestions for our experiment.
We would be very grateful if the revised manuscript could be reviewed and considered for publication in Cells. Everything goes well!!
Reviewer 2 Report
The current study investigated the neuroprotective effect of abrocitinib on neuroinflammation after TBI, they found that abrocitinib reduce excessive neuroinflammation by restraining the JAK1/ 45 STAT1 / NF-κB pathway. They included both in vivo and in vitro studies. It is an interesting study but there are several issues that are needed to be considered:
1. How many animals are used in each group?
2. Which sex included in this study?
3. How old were the mice?
4. It is important to have another group with only abrocitinib to see the effect of abrocitinibin normal condition.
5. Which part of brain is selected for RNAseq?
6. It is not clear RNAseq was performed for mice with TBI+ abrocitinibin or not?
7. How the weight of mice was changed at different time points after surgery (1st, 3rd, and 7th days after FPI)?
8. Did MRI perform on the same mice that used for RNAseq?
9. What is the reason of using only one time point (day 1) for Nissl and H&E staining?
10. How many animal were used for each group and each experiment?
11. Authors mentioned “Frozen 6-μm-thick tissue sections were prepared as described above”, however, the previous preparation was about paraffin sectioning, so it is important to explain how frozen tissue prepared?
12. How many sections were used for each staining?
13. It is not clear how both (ANOVA) test and nonparametric test were used to compare the significant different among groups?
14. Did authors check normal distribution of data? And homogeneity of data? What was the post hoc test?
15. It is not clear if different days are on the same mice or different mice? If same mice the repeated measure ANOVA is needed? If not then there is another independent factor which is the day and therefore two way ANOVA is needed to apply.
16. In figure 3 , it is not clear that images are from which brain region?
17. In fig6.B what is the staining? And which brain region?
Author Response
Thank you for your recognition and suggestions for our experiment.
We have revised the manuscript according to your review comments. Further, we embellished the language for the convenience of readers. The details could be found in the manuscript and attachments.
We would be very grateful if the revised manuscript could be reviewed and considered for publication in Cells. Everything goes well!!

Round 2
Reviewer 2 Report
Authors responded to the comments in a satisfactory manner.